# Psychological Traits and Behavioural Influences in Patients with Dystonia—An Observational Cohort Study in a Romanian Neurology Department

**DOI:** 10.3390/life11070612

**Published:** 2021-06-24

**Authors:** Eugenia Irene Davidescu, Irina Odajiu, Delia Tulbă, Iulia Mitrea, Camelia Cucu, Bogdan Ovidiu Popescu

**Affiliations:** 1Department of Clinical Neurosciences “Carol Davila”, University of Medicine and Pharmacy, 030167 Bucharest, Romania; eugenia.davidescu@umfcd.ro (E.I.D.); delia.tulba@umfcd.ro (D.T.); iulia.popescu-olaru@drd.umfcd.ro (I.M.); 2Department of Neurology, Colentina Clinical Hospital, 020125 Bucharest, Romania; irina.odajiu@rez.umfcd.ro (I.O.); c.d.cucu@gmail.com (C.C.); 3Colentina—Research and Development Center, Colentina Clinical Hospital, 020125 Bucharest, Romania; 4Department of Cell Biology, Neurosciences and Experimental Myology, “Victor Babeș”, National Institute of Pathology, 050096 Bucharest, Romania

**Keywords:** dystonia, personality profile, DECAS, openness

## Abstract

(1) Background: Emerging evidence indicates that non-motor symptoms significantly influence the quality of life in dystonic patients. Therefore, it is essential to evaluate their psychological characteristics and personality traits. (2) Methods: Subjects with idiopathic dystonia and a matched control group were enrolled in this prospective observational cohort study. Inclusion criteria for patient group included idiopathic dystonia diagnosis, evolution exceeding 1 year, and signed informed consent. Inclusion criteria for the control group included lack of neurological comorbidities and signed informed consent. All subjects completed the DECAS Personality Inventory along with an additional form of demographic factors. Data (including descriptive statistics and univariate and multivariate analysis) were analyzed with SPSS. (3) Results: In total, 95 participants were included, of which 57 were in the patient group. Females prevailed (80%), and the mean age was 54.64 ± 12.8 years. The most frequent clinical features of dystonia were focal distribution (71.9%) and progressive disease course (94.73%). The patients underwent regular treatment with botulinum toxin (85.95%). In addition, patients with dystonia obtained significantly higher openness scores than controls, even after adjusting for possible confounders (*p* = 0.006). Personality traits were also different between the two groups, with patients more often being fantasists (*p* = 0.007), experimenters (*p* = 0.022), sophists (*p* = 0.040), seldom acceptors (*p* = 0.022), and pragmatics (*p* = 0.022) than control subjects. (4) Conclusion: Dystonic patients tend to have different personality profiles compared to control subjects, which should be taken into consideration by the treating neurologist.

## 1. Introduction

Dystonia is a movement disorder presenting with sustained/intermittent muscle contractions that determine abnormal, repetitive postures and/or movements. It is often induced or worsened by voluntary action coupled with excessive muscle activation [1]. Dystonia often presents a focal body distribution, which is 10-times more common than the generalized form [2]. The prevalence of idiopathic dystonia is around 16.4 per 100,000 people [3].

Similar to other movement disorders such as Parkinson’s or Huntington’s disease, there is increasing evidence that patients with dystonia experience non-motor symptoms. Among these, mood alterations—of which depression and anxiety are the most frequent—cognitive impairment, sleep problems, pain, and dysautonomia are also worth mentioning [4], mainly because of their negative impact on quality of life. The prevalence of depression and anxiety in patients with dystonia ranges between 12% and 71% throughout life [4,5,6]. Since these disturbances can manifest before the onset of dystonia, they could be part of the disease [7] and not just reactive to the motor symptoms. Nevertheless, there are still conflicting data on whether mood alterations are secondary to motor manifestations and consecutive psychosocial impairment or primary manifestations of dystonia. Interestingly, carriers of the DYT1 gene mutation have a consistently increased rate and an earlier onset of major depressive disorder than noncarrier controls [8]. Notably, no correlation between depression/anxiety and motor symptom severity has been identified so far [9], Moreover, psychiatric manifestations seem to be more significant predictors of health-related quality of life than dystonia severity [10].

Apart from modulating the susceptibility to psychiatric manifestations, certain personality traits could also have an important influence on dystonia, provided that they shape the way these patients react to both internal and external changes [11]. In line with Eysenck’s definition of personality—“habitual patterns of behavior, and behavior, of course, can be modified”—it has been suggested that dystonic patients, like other patients with chronic disabling diseases, are inclined to develop higher neuroticism, lower extraversion, and higher anxiety and obsessionality levels [12,13]. However, the evidence involving personality changes in dystonic patients is rather heterogeneous. For example, some early studies found no evidence of abnormal personality structure in torticollis patients [14,15], whereas other articles have reported a tendency for neurotic personality traits [16,17].

It was also noticed that dystonic patients have an increased rate of anxious personality disorders (32.6%), specifically obsessive-compulsive and avoidant personality disorders [18]. According to a large cohort study, 19.7% of patients with idiopathic focal dystonia fulfilled the diagnostic criteria for the obsessive-compulsive disorder [19]. Additionally, symptomatic carriers of the DYT11 mutation (characteristic for myoclonus-dystonia) showed an increased rate of obsessive-compulsive symptoms compared to nonsymptomatic ones and the control population. Furthermore, they were more prone to develop alcohol dependence [20]. Patients with dystonia are more inclined to higher rates of emotional instability along with a reduced emotional control, increased suspiciousness and frustration, low mood levels, and reduced resistance to stress [21]. They have low self-esteem, isolate themselves [22], and develop social phobia [5,18]. A study found that nonefficient coping strategies in patients with spasmodic torticollis correlated with an increased rate of psychiatric diagnoses, suggesting that the high mental disorder burden in these patients is not merely a consequence of chronic disease and disfigurement [23].

Apart from the motor symptoms, the evaluation of dystonic patients should also include psychological characteristics and personality traits. This approach might improve the doctor-patient relationship and guide it toward adherence to treatment and early detection of concurrent psychiatric comorbidities that could benefit from symptomatic treatment. This study aimed to identify whether patients with idiopathic dystonia exhibit peculiar, different personality profiles compared to a matched control group.

## 2. Materials and Methods

We conducted a unicentric prospective observational cohort study involving patients with dystonia monitored in the outpatient section of the Neurology Department of Colentina Clinical Hospital from Bucharest, Romania, a reference center for dystonic patients, along with a control group of volunteer subjects with no neurological diseases. Inclusion criteria for the patient group included an established diagnosis of idiopathic dystonia (according to the Guidelines for the diagnosis and treatment of primary (idiopathic) dystonia reported by an EFNS MDS-ES Task Force), an evolution exceeding 1 year, and signed consent to participate in the study. Inclusion criteria for the control group included no neurological comorbidity and signed informed consent. The recruitment phase of the participants took place between 19–31 December 2019.

Informed consent was obtained from all the participants. The study was approved by the local Ethics Committee (No. 13/17 December 2019) and was performed according to the World Medical Association Declaration of Helsinki from 1975.

All the participants individually completed the DECAS Personality Inventory [24], comprising 97 questions that employ dichotomic (true/false) answers to items uniformly assigned to the 5 fundamental dimensions of personality: Openness, Extraversion, Conscientiousness, Agreeability, and Emotional stability. Data were introduced and processed using a digital tool that automatically calculated the scores and assigned the personality dimension and traits (based on combinations of personality dimensions).

Additionally, the participants completed an ad-hoc questionnaire with questions regarding age, sex, dominant hand, alcohol and tobacco consumption, studies, occupation, religion, and marital status. The questionnaires were handled by a member of our research team who also inquired about family history and recorded dystonia characteristics such as type of dystonia, disease duration, body distribution, disease course and variability, and type of the administered treatment.

Database design and data analysis were performed using IBM SPSS Statistics 25. Categorical variables were reported as frequency and analyzed with a Chi-square test. Normality was tested with Kolmogorov–Smirnov test. Continuous variables not normally distributed were reported as median (minimum, maximum) and analyzed with the Mann–Whitney U test. Spearman rank-order correlation coefficient was used to measure the strength and direction of association between continuous variables. In the linear regression, we adjusted for all the variables that significantly correlated with the personality dimensions, which differed between the 2 groups (*p* < 0.05). The possible confounders were selected as Independent(s) by the Enter method, with the scales assessing personality dimensions as the Dependent variable. Hypothesis testing was 2-tailed and statistical significance was defined as *p* < 0.05.

## 3. Results

In total, 95 participants were included in the study, all of Romanian nationality, of which 57 were in the patient group. The mean age was 54.64 ± 12.8 years, ranging between 22 and 76 years, and a more significant proportion were females (80%). The control and the patient groups were similar in terms of mean age (54.39 years vs. 54.81) and sex distribution (women prevailed in both groups: 81.57% vs. 78.94%). None of the patients had familial clustering.

Demographic factors were similar in the two groups (Table 1). Married, orthodox, retired subjects with secondary education, right-hand dominance, and no history of alcohol or tobacco consumption prevailed in both groups.

In the patient group, the mean disease duration was 7.25 years (±7.83), and focal dystonia was the most prevalent (71.9%), mostly cervical dystonia (49.12%). Most of the patients underwent regular treatment with botulinum toxin within the last year (85.95%), whereas 24.56% also took clonazepam and trihexyphenidyl. The majority had progressive disease (94.73%) and diurnal fluctuations (61.4%) (Table 2).

Compared to the control group, the subjects from the patient group obtained a higher median score in all personality dimensions of the DECAS Personality Inventory (Table 3), except for emotional stability. However, openness was the only personality dimension significantly associated with dystonia (*p* = 0.020). In the univariate analysis of the patient group, younger subjects, smokers, heavy drinkers, and extroverts had greater openness (*p* = 0.035, *p* = 0.009, *p* = 0.015, and *p* < 0.001, respectively). Multiple regression was performed to predict openness from dystonia and all these possible confounders. After adjusting for all of them, openness remained significantly associated with dystonia (*p* = 0.006) and younger age (*p* = 0.041) (Table 4).

As expected, in the patient group, higher extraversion scores were associated with younger age (*p* = 0.001), male sex (*p* = 0.049), smoking (*p* = 0.005), and alcohol consumption (*p* = 0.002).

Conscientiousness was significantly associated with emotional stability (*p* = 0.009, the more emotionally stable patients were, the less conscientious they seemed). Patients with progressive disease course were less emotionally stable (*p* = 0.023). On the other hand, the more emotionally stable the patients were, the higher levels of agreeability they presented (*p* = 0.001).

Concerning the personality profile [24], openness is a determinant of the subject’s resistance to stress (along with emotional stability), cognitive style (along with conscientiousness), orientation in life (along with extraversion), and orientation in society (along with agreeability). Compared to the control group, the patients with dystonia were significantly less often acceptors (*p* = 0.022) and pragmatics (*p* = 0.022) and more frequent fantasists (*p* = 0.007), experimenters (*p* = 0.022), and sophists (*p* = 0.040) (Table 5). From all the personality traits, dystonia severity was significantly associated only with an avoidant profile (*p* = 0.023), meaning that patients with segmental or multifocal dystonia were more often avoidant than those with focal distribution.

## 4. Discussion

According to the results of this prospective observational cohort study, patients with idiopathic dystonia presented more openness than control subjects (*p* = 0.020). This personality dimension is an essential determinant of patients’ resistance to stress, cognitive style, orientation in life, and society. Furthermore, in the patient group, openness was positively correlated with extraversion (*p* < 0.001). Interestingly, a great majority of dystonic patients did not have alcohol or tobacco addiction, but the smokers and heavy drinkers within this group had higher extraversion and openness scores.

Regarding stress resistance, dystonic patients were more often fantasists (*p* = 0.007) and less frequently acceptors (*p* = 0.022). Additionally, as expected, patients with segmental or multifocal distribution of dystonia were more often avoidant than those with focal distribution. Concerning the dimension of orientation in life, the dystonic patients were often experimenters (43.85%) (*p* = 0.022), meaning that they prefer innovative approaches and new experiences but are prone to get bored quickly and renounce practicing them. This could negatively influence their treatment adherence, as they could become noncompliant to a new treatment after a while. On the other hand, they were less frequent pragmatic (*p* = 0.022) than those in the control group. Finally, regarding orientation in society, dystonic patients were more often sophists (*p* = 0.040), preferring to have a conflicting opinion compared to others and to debate all problems critically without being affectively involved. Dystonic patients were also often progressists (28.07%), open-minded, tolerant, and encouraged opinion diversity. These results demonstrate that patients with dystonia have distinct personality profiles compared to the control group. We also detected a higher unemployment percentage in dystonic patients, mainly because of earlier retirement.

Our result for openness comes in opposition to the findings of another study conducted by Lencer et al. [18], where dystonic patients presented lower values for this dimension. Still, its positive correlation with extraversion supports the results from another study that enrolled patients with primary focal dystonia [8]. Interestingly, subjects from other studies more often had tobacco or alcohol addictions [20,25] compared to our results.

By having fantasist (38.59%) and avoidant (33.3%) personality traits, patients prefer to isolate themselves from the real world, create an imaginary one, and exaggerate things. These findings align with those of another study that enrolled patients with spasmodic torticollis, indicating that patients with dystonia have high rates of social phobia [5]. This high prevalence could be due to a “maladaptive attitude” toward the illness and body image and not necessarily related to the severity of dystonia [5]. Interestingly, anxiety and social phobias occur more often in musicians with dystonia, as they tend to have excessive perfectionist tendencies [26]. However, these traits seem to be pre-existent and not psychoreactive phenomena [26]. The development of secondary social phobia in patients with idiopathic focal dystonia could be enhanced by personality traits and avoidance of social activities because of difficulties in coping with dystonia symptoms [18].

Our results about the early retirement rate are supported by evidence from other studies [27,28]. For example, another study involving 1038 patients with cervical dystonia found that 38.5% of unemployed subjects lost their job because of their condition. Possible explanations reside from the effect of the disease itself on patients’ productivity and stigmatization. According to a cross-cultural comparison study, patients with generalized dystonia are subjected to a relatively high percentage of stigma (33%) [29], which leads to increased psychological distress, poorer health outcomes [30], and negative attitudes toward work [31].

Concerning cognitive style, the patient group frequently exhibited a refractory attitude (36.84%). In other words, they prefer simple tasks, manifest resistance toward learning new things and are often chaotic, and do not respect deadlines. These results are in line with the results of other studies, which showed that patients with dystonia present mild executive dysfunction related to set-shifting deficits, verbal learning, category fluency, and dual-task performance [32,33].

The divergence of our findings compared to other studies could be related to the heterogeneity of dystonia types, since we did not specifically select patients with a particular type of dystonia, as opposed to previously mentioned studies that addressed personality traits in spasmodic torticollis or musician’s dystonia. Another reason for the conflicting results could reside from different nationality and demographic backgrounds in our patient group. Nevertheless, studies with a greater sample size with different cultural backgrounds and dystonia types are needed to establish the relationship between personality, nationality, and dystonia type.

The predominance of certain personality traits and psychologic manifestations in these patients could either be a consequence of the pathophysiology of dystonia or its negative influence on the perception of body image, which might cause functional disability [34]. We emphasize the fact that patients with dystonia seem to have abnormal striato-frontal circuits, which are also involved in regulating mood and behavior [35,36]. Other loops with abnormal functioning supposedly linked to the non-motor symptoms of dystonia relate to the cortex, limbic system, basal ganglia, and cerebellum [34]. For instance, in primary focal dystonia, the impaired activity of the motor loops that link the basal ganglia to the frontal cortex could also influence the limbic loops, leading to alteration in motor and affective processing [37]. Elevated perseverance, which is frequently found in focal dystonia, displays a pattern of complex neuropsychological dysfunction that involves the dorso-lateral, orbito-frontal, and motor fronto-striatal circuits [32].

The main limitations of our study relate to the small sample size and the lack of an objective measure of personality and behavior traits meant to ascertain that patients perceive themselves factually. The female prevalence (80%) in our patient group, another limitation to be considered, could be related to the fact that focal cranial and cervical dystonia types prevailed in our patients, which affect females more often than males [38].

Nevertheless, the small number of variables required in the multivariate analysis justify the statement that dystonia and age are independently associated with higher openness (regardless of sex, smoking, and alcohol consumption), even though the patient group is small. Indeed, it is debatable whether modified personality profiles are part of the spectrum of non-motor symptoms in movement disorders, mainly because they do not have a clear meaning in neurology. However, since personality traits essentially influence certain domains of the non-motor symptoms such as fatigue/sleep, mood/cognition, and attention/memory, it may not be hazardous to assess them in this setting. Yet, their implications (both causal and consequential) in non-motor manifestations of dystonia need further investigation.

Since there is a lack of curative or disease-modifying treatment options for dystonia, it is essential to establish a trustworthy long-term therapeutic alliance with the patient. Identifying patients’ personality traits might help increase the compliance to treatment and ascertain which patients are prone to be excessively influenced by the disease and would benefit from psychotherapy.

## 5. Conclusions

Our study proves that dystonic patients tend to have significantly different personality profiles compared to other subjects. They are less often acceptors and more often experimentalists and sophists, even though they are more open to trying new things. They renounce easily after adhering to new practices and prefer to have a conflicting opinion from others. Knowing their particular personality traits and profiles might help the neurologist adjust the therapeutic approach and increase their compliance to therapy.

## Figures and Tables

**Table 1 life-11-00612-t001:** Demographic characteristics.

	Patient Group (No. (%))	Control Group (No. (%))	*p* Value
**Education:**			0.180
– 8 classes	2 (3.5%)	3 (7.89%)	
– 10 classes	5 (8.77%)	5 (13.15%)	
– Vocational school	3 (5.26%)	2 (5.26%)	
– High school	18 (31.57%)	19 (50%)	
– Post-secondary school	5 (8.77%)	1 (2.63%)	
– University	24 (42.10%)	8 (21.05%)	
**Dominant Hand:**			0.792
– Right	53 (92.98%)	36 (94.73%)	
– Left	3 (5.26%)	1 (2.63%)	
– Ambidextrous	1 (1.75%)	1 (2.63%)	
**Occupation:**			0.196
– Retired	26 (47.36%)	16 (42.10%)	
– None	6 (18.42%)	1 (2.63%)	
– Middle class	21 (36.84%)	15 (39.47%)	
– Working-class	4 (7.01%)	6 (15.78%)	
**Religion:**			0.408
– Atheist	1 (1.75%)	2 (5.26%)	
– Orthodox	53 (92.98%)	36 (94.73%)	
– Catholic	2 (3.5%)	0	
– Lutheran	1 (1.75%)	0	
**Marital Status:**			0.565
– Single	12 (21.05%)	6 (15.78%)	
– Married	36 (63.15%)	29 (76.31%)	
– Divorced	3 (5.26%)	1 (2.63%)	
– Widow	6 (10.52%)	2 (5.26%)	
**Alcohol Consumption:**			0.095
– None	53 (92.98%)	33 (86.84%)	
– Occasional	4 (7.01%)	2 (5.26%)	
– Regular	0	3 (7.89%)	
**Smoking:**			0.146
– Non-smoker	50 (87.71%)	29 (76.31%)	
– Smoker	7 (12.28%)	9 (23.68%)	

**Table 2 life-11-00612-t002:** Characteristics of dystonia.

**Disease Duration, Years (Mean, SD, Range)**	**7.25 ± 7.83 (1–35)**
**Dystonia Type**	
– Bilateral blepharospasm	14 (24.56%)
– Cervical dystonia *	28 (49.12%)
– Complex dystonia	15 (26.31%)
**Body Distribution**	
– Focal	41 (71.9%)
– Segmental	16 (28.1%)
– Multifocal	0 (0%)
– Generalized	0 (0%)
**Disease Course**	
– Static	3 (5.26%)
– Progressive	54 (94.73%)
**Variability**	
– Persistent	22 (38.6%)
– Action specific	0 (0%)
– Diurnal fluctuations	35 (61.4%)
– Paroxysmal	0 (%)
**Treatment with Botulinum Toxin in the Previous Year**	
– Yes	49 (85.95%)
**Oral Treatment in the Previous Year**	
– No	24 (42.1%)
– Trihexyphenidyl	5 (8.77%)
– Clonazepam	14 (24.56%)
– Both	14 (24.56%)

* spasmodic torticollis, laterocollis, retrocollis.

**Table 3 life-11-00612-t003:** Scores for the DECAS questionnaire.

	Openness	Extraversion	Conscientiousness	Agreeability	Emotional Stability
Patient Group	51.5	50.5	45	47.4	44.4
	(32.4–73.2)	(26.7–80)	(29.4–80)	(26.7–67.5)	(26.7–67.5)
Control Group	46.1	49.3	42.4	46.1	44.4
	(26.7–70.5)	(33.5–80)	(26.7–62.2)	(29.4–80)	(20–67.5)
*p* Value	0.020	0.569	0.338	0.635	0.405

**Table 4 life-11-00612-t004:** Dystonia and age are independently associated with openness, adjusted for sex, smoking, and alcohol consumption (linear regression).

Variables	Unstandardized Coefficients B Standard Error		95% CI for B Lower Bound Upper Bound		*p* Value
Dystonia	5.665	2.002	1.686	9.643	0.006
Age	−0.170	0.082	−0.332	−0.007	0.041
Sex	0.193	2.558	−2.134	8.030	0.252
Smoking	0.193	2.842	−5.454	5.840	0.946
Alcohol consumption	1.572	2.518	−3.432	6.576	0.534

**Table 5 life-11-00612-t005:** Personality profile.

Personality Traits	Control Group	Patient Group	*p* Value
**Resistance to Stress:**			
– Acceptor	11 (28.94%)	6 (10.5%)	0.022
– Fantasist	5 (14.28%)	22 (38.59%)	0.007
– Avoidant	17 (44.73%)	19 (33.3%)	0.262
– Rationalist	5 (14.28%)	10 (17.54%)	0.566
**Cognitive Style:**			
– Analyst	4 (10.52%)	15 (26.31%)	0.059
– Conformist	7 (18.42%)	4 (7.01%)	0.089
– Designer	6 (15.78%)	17 (29.82%)	0.118
– Refractory	21 (55.26%)	21 (36.84%)	0.077
**Orientation in Life:**			
– Domestic	19 (50%)	20 (35.08%)	0.148
– Experimenter	8 (21.05%)	25 (43.85%)	0.022
– Philosopher	1 (2.53%)	7 (12.28%)	0.097
– Pragmatic	10 (26.31%)	5 (8.77%)	0.022
**Orientation in Society:**			
– Fundamentalist	17 (44.73%)	15 (26.31%)	0.063
– Progressionist	5 (13.15%)	16 (28.07%)	0.086
– Sophist	4 (10.52%)	16 (28.07%)	0.040
– Traditionalist	12 (31.57%)	10 (17.54%)	0.112

## Data Availability

The raw data supporting the conclusions of this article will be made available by the authors without undue reservation.

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
