# Peer review of "Psychological Traits and Behavioural Influences in Patients with Dystonia—An Observational Cohort Study in a Romanian Neurology Department"

_life, 2021, doi:10.3390/life11070612_

Round 1
Reviewer 1 Report
In their manuscript "Psychological traits and behavioural influences in patients with dystonia - an observational cohort study in a Romanian Neurology Department" Davidescu et al. analyze non-motor symptoms of patients with dystonia and place their findings in the context of the existing study literature.
The article is well written and includes references to the most important studies published in the field of dystonia.
General Comments:
The article describes well the demographic composition of the patient and control groups. Here, a clear preponderance of female subjects within the patient group is noticeable. This is mentioned in the article, but an explanatory part is missing in the discussion, which takes a closer look at this strong surplus of female participants.
While in the control group young patients were described as more open, in the patient group a positive correlation of age and openness is found. This apparent discrepancy should also be examined more closely by the authors.
In my opinion, the group of dystonia patients presented here is not classified enough by type of dystonia. It would be expected that mild and more focal dystonia would have a lesser impact on psychological well-being, and thus on behaviors, than more severe courses. The authors should again include the severity of dystonia in their analyses and evaluate it as a variable.
Minor comments:
The article should be rechecked for grammatical errors and incorrect punctuation.
Author Response
We do thank the reviewer for time and expertise spent in order to improve our article. We addressed all issues raised by the reviewer, as can be seen point by point below.
General Comments:
The article describes well the demographic composition of the patient and control groups. Here, a clear preponderance of female subjects within the patient group is noticeable. This is mentioned in the article, but an explanatory part is missing in the discussion, which takes a closer look at this strong surplus of female participants.
This issue was commented in the Discussion section in lines 251-253.
While in the control group young patients were described as more open, in the patient group a positive correlation of age and openness is found. This apparent discrepancy should also be examined more closely by the authors.
Actually, there is a significantly negative association between openness and age (B coefficient=-0.170)- we clarified this in lines 150-151.
In my opinion, the group of dystonia patients presented here is not classified enough by type of dystonia. It would be expected that mild and more focal dystonia would have a lesser impact on psychological well-being, and thus on behaviors, than more severe courses. The authors should again include the severity of dystonia in their analyses and evaluate it as a variable.
This was addressed in the Results section – lines 169-171 and in the Discussion section lines 183-185.
Minor comments:
The article should be rechecked for grammatical errors and incorrect punctuation.
We did our best to improve grammar and punctuation.
Reviewer 2 Report
Review report
Comments to the authors
General comments
In general, the manuscript is well written and easy to read. Its subject is interesting and it will contribute to possible improvements of clinical practice as it provides interesting information towards the current knowledge. The authors need to re-organise the information in the methods section, improve the abstract and modify some minor details. Please find below the minor changes to address.
Specific comments
The following minor points should be addressed by the authors.
Please indicate in your response to the comments in which line of the manuscript you have made changes.
Abstract
The inclusion criteria should be specified in the methods section of the abstract.
The conclusion should include what for the ‘particular personality profiles which should be taken into account’. The sentence seems incomplete.
Introduction
Line 44: references 4,5,6 should be included inside the same [..] i.e: [4,5,6]. The same appears in other lines in the manuscript, please change.
Methods
The inclusion criteria are not clear, different criteria are mixed under the same sentence. Please indicate inclusion criteria separated by comas.
Please write the methods section in chronological order. For example, I think that the recruitment of the participants was done before completing the questionnaire and before they signed the informed consent, so please introduce the phrase regarding the recruitment before the explanation about the complexion of the questionnaires and the informed consent. Reorganised the methods section according to chronological order to facilitate understanding the procedure.
When the authors say that ‘the participants completed a form with questions’ I think it would be more appropriate to call that form an ad-hoc questionnaire.
The authors should include specify in the manuscript who handled the questionnaires and the data, was it a member of the research team? Was it an independent researcher?
Discussion
The first paragraph of the discussion should summarise the objective of the study and the more important findings of the study. After this information the comparison of results should start.
Lone 172: please specify the author of the other study to complete the sentence: …another study conducted by… and even explain more about that study.
Line 195: please change dystonia at the end of the sentence for ‘their condition’.
Author Response
We do thank the reviewer for the time and expertise and we were happy to adjust the manuscript according to the issues raised, as described below.
General Comments:
The article describes well the demographic composition of the patient and control groups. Here, a clear preponderance of female subjects within the patient group is noticeable. This is mentioned in the article, but an explanatory part is missing in the discussion, which takes a closer look at this strong surplus of female participants.
This issue was commented in the Discussion section in lines 251-253.
While in the control group young patients were described as more open, in the patient group a positive correlation of age and openness is found. This apparent discrepancy should also be examined more closely by the authors.
Actually, there is a significantly negative association between openness and age (B coefficient=-0.170)- we clarified this in lines 150-151.
In my opinion, the group of dystonia patients presented here is not classified enough by type of dystonia. It would be expected that mild and more focal dystonia would have a lesser impact on psychological well-being, and thus on behaviors, than more severe courses. The authors should again include the severity of dystonia in their analyses and evaluate it as a variable.
This was addressed in the Results section – lines 169-171 and in the Discussion section lines 183-185.
Minor comments:
The article should be rechecked for grammatical errors and incorrect punctuation.
We did our best to improve grammar and punctuation.
Specific comments
The following minor points should be addressed by the authors.
Please indicate in your response to the comments in which line of the manuscript you have made changes.
Abstract
The inclusion criteria should be specified in the methods section of the abstract.
This was included in lines 18-20.
The conclusion should include what for the ‘particular personality profiles which should be taken into account’. The sentence seems incomplete.
This issue was addressed in lines 30-31.
Introduction
Line 44: references 4,5,6 should be included inside the same [..] i.e: [4,5,6]. The same appears in other lines in the manuscript, please change.
Thank you for the suggestion, all references have been corrected accordingly.
Methods
The inclusion criteria are not clear, different criteria are mixed under the same sentence. Please indicate inclusion criteria separated by comas.
We have addressed this issue in lines 93-95.
Please write the methods section in chronological order. For example, I think that the recruitment of the participants was done before completing the questionnaire and before they signed the informed consent, so please introduce the phrase regarding the recruitment before the explanation about the complexion of the questionnaires and the informed consent. Reorganised the methods section according to chronological order to facilitate understanding the procedure.
Thank you for the suggestion the order has been modified accordingly, please note lines 95-100.
When the authors say that ‘the participants completed a form with questions’ I think it would be more appropriate to call that form an ad-hoc questionnaire.
Thank you for the suggestion, we added the term in line 107.
The authors should include specify in the manuscript who handled the questionnaires and the data, was it a member of the research team? Was it an independent researcher?
We included this information (lines 109-110).
Discussion
The first paragraph of the discussion should summarise the objective of the study and the more important findings of the study. After this information the comparison of results should start.
Thank you very much for the suggestion, the discussion section was modified accordingly.
Lone 172: please specify the author of the other study to complete the sentence: …another study conducted by… and even explain more about that study.
We have addressed this issue in line 199.
Line 195: please change dystonia at the end of the sentence for ‘their condition’.
Thank you for the suggestion we have made the modification in line 218.